# Fetal Congenital Complete Heart Block: A Rare Case with an Extremely Low Ventricular Rate and Review of Current Management Strategies

**DOI:** 10.3390/children10071132

**Published:** 2023-06-29

**Authors:** Stefani Samples, Catherine Fitt, Michael Satzer, Ronald Wakai, Janette Strasburger, Sheetal Patel

**Affiliations:** 1Pediatric Cardiology, Ann & Robert Lurie Children’s Hospital of Chicago, Chicago, IL 60611, USA; 2Department of Medical Physics, University of Wisconsin, Madison, WI 53705, USA; 3Pediatric Cardiology, Children’s Hospital of Wisconsin, Milwaukee, WI 53226, USA

**Keywords:** congenital complete heart block, fetal bradycardia, autoimmune-mediated heart block

## Abstract

Congenital complete heart block (CCHB) is associated with high intrauterine and post-natal mortality. Prenatal detection and management, as well as appropriate delivery planning, may improve the outcomes in CCHB. We describe a rare case of CCHB that initially presented with fetal ascites and high-grade second-degree heart block noted on fetal echocardiography. The mother was noted to be positive for anti-SSA antibodies, and treatment with maternal steroids was started in an effort to reverse the fetal cardiac conduction abnormality. However, the fetal cardiac rhythm progressed to complete heart block by the follow up evaluation and the fetus had a continual declination of heart rate throughout the pregnancy to a low fetal heart rate of 25 beats per minute (bpm). This case demonstrates the lowest fetal ventricular rate documented in the literature and illustrates a severe presentation of a rare disease process. An overview of the existing knowledge related to etiology, prenatal evaluation with fetal echocardiography and fetal magnetocardiography, prenatal management, and delivery planning in fetuses with prenatally detected CCHB is included.

## 1. Introduction

The following case presentation demonstrates a severe presentation of a rare disease process: congenital complete heart block (CCHB). CCHB may occur in association with maternal autoimmune disease or with structural congenital heart disease. The diagnosis of maternal autoimmune disease is often not made until after the diagnosis of fetal CCHB, and significant morbidity and mortality remain despite advancing treatment options.

## 2. Case Presentation

A 34-year-old G5P4 woman was referred to our fetal health institute at 23.3 weeks gestation for evaluation including a fetal echocardiogram due to fetal ascites noted approximately ten days prior to initial presentation. Her previous prenatal care was performed by a midwife, and the ascites was noted on a limited screening ultrasound with a documented fetal heart rate of 136 bpm at that time. Her past obstetrical history was uncomplicated. Her medical history was only remarkable for an immune thrombocytopenic purpura diagnosis approximately 15 years prior, which resolved after treatment with prednisone and rituximab. Since that time, she reported no major medical issues, and her only regular medication was prenatal vitamins.

On initial fetal echocardiogram, findings were significant for fetal arrhythmia with intermittent high-grade 2nd-degree heart block alternating with 1:1 atrioventricular (AV) conduction for a few beats and ventricular rates ranging from 58 to 126 bpm (Figure 1). Significant ascites, cardiomegaly, and increased echogenicity of multiple intracardiac structures including the endocardium (Figure 2) raised suspicion for an autoimmune-related process. Due to concern for developing complete heart block in the setting of presumed maternal autoimmune disease, maternal steroid therapy with dexamethasone (8 mg PO once daily) was started while awaiting serology. Lab results confirmed anti-ssA (Ro) positivity. Despite the initiation of high-dose steroid therapy, the fetus progressed from high 2nd degree heart block to complete heart block with a ventricular rate of 53 bpm by the follow-up fetal echocardiogram five days later (24.0 weeks gestation). Terbutaline therapy was offered with the goal of increasing the fetal heart rate, though this was declined by the patient due to concern for potential maternal side effects. The fetal heart rhythm was further confirmed by fetal magnetocardiography (fMCG) at 24.3 weeks gestation with an average fetal ventricular rate of 38–41 bpm (Figure 3). Cardiovascular profile score (CVP) and the ventricular rate continued to decrease, with the lowest heart rate measurement of 25 bpm at 32.0 weeks gestation (Figure 1). Due to declining CVP and high clinical concern for intrauterine fetal demise, a multidisciplinary team meeting was conducted with the parents to discuss the option of early delivery via caesarean section and postnatal plans including prompt neonatal resuscitation and urgent cardiac intervention. Planned postnatal cardiac intervention was to include external pacing by surgically placing epicardial pacing wires via sternotomy and using a temporary external pacemaker device until the infant was of sufficient size to allow permanent pacemaker device placement in the abdominal wall. Potential risks of prematurity in the setting of a hydropic newborn were also discussed to arrive at shared decision-making. The family declined emergent cesarian section delivery and decided to wait and observe. In utero fetal demise was noted one week later at 33 weeks gestation.

## 3. Discussion

Congenital complete heart block (CCHB) is a rare entity, with a reported incidence of 1 in 15,000 to 1 in 20,000 births [1].

### 3.1. Etiology and Expected Outcomes

CCHB may occur in association with maternal autoimmune disease or with structural congenital heart disease. Structural congenital heart diseases most commonly associated with CCHB includes left atrial isomerism (polysplenia), L-transposition of the great arteries, or atrioventricular septal defect. When associated with congenital heart disease, CCHB has a poorer prognosis, with one prior study reporting fetal mortality with intrauterine hydrops and fetal demise in 7% of their cohort, with an additional 10–15% mortality during infancy alone [2]. Compared to autoimmune-mediated complete heart block, the fetuses with structural CHD and heart block often demonstrate non-reactive fetal heart rate tracings on fMCG and less response to treatment with terbutaline [3,4].

Autoimmune-mediated CCHB occurs with normal fetal cardiac anatomy but with the presence of maternal ssA (Ro) or ssB (La) antibodies. CCHB in these cases is thought to result from an inflammatory response in the fetal heart from the transplacental transfer of the autoantibodies, which causes tissue injury, fibrosis, and/or scarring of the cardiac conduction system [5]. This inflammation occurs most commonly between 18 and 25 weeks of gestation, though the diagnosis may not be made until after birth in some cases [6]. Mothers with these antibodies have a risk of fetal CCHB development in 1–2% of cases; however, this risk rises to 16–40% in subsequent pregnancies if a previous child has CCHB [2]. While maternal antibodies are detectable in 95% of mothers of infants with isolated CCHB, only 20–30% of infants with CCHB have mothers with a previous diagnosis of autoimmune disease, and a majority are asymptomatic [5]. As in our case, this lack of pre-existing maternal symptoms further complicates the appropriate prenatal diagnosis of this rare disease.

While outcomes for isolated CCHB are not as poor as those associated with congenital heart disease, morbidity and mortality remain significant both prenatally and in the neonatal period. In a multicenter- retrospective study, perinatal morbidity and mortality were 58% and 7%, respectively, with a total mortality of 16% during the 50 years studied, 73% of which occurred in the first 12 months of life [7]. Prematurity also confers additional risk with complications of prematurity and more frequent need for inotropic support and temporary pacing while awaiting permanent pacemaker placement [8]. Poorer overall perinatal outcomes have been associated with lower ventricular rates < 50 bpm, rapidly decreasing fetal heart rate, and other findings of poor ventricular function, atrioventricular valve regurgitation, or hydrops [2]. Low ventricular rates of 37 bpm, 38 bpm, and 43 bpm have been reported in cases of immune-mediated fetal heart block, making our case the lowest documented fetal heart rate in the literature at 25 bpm [9,10]. These cases, along with ours, demonstrate the possibility that the fetus can survive for some period of time despite very low ventricular rates. With close clinical follow up, this may provide time for interventions to maximize fetal survival where able, as well as consideration for early delivery with postnatal pacing as fetal weight and gestational age allow.

### 3.2. Prenatal Cardiac Evaluation

Diagnosis of CCHB is generally made by fetal echocardiography using Doppler echocardiography and M-mode interrogation of the atrial and ventricular wall motion. Pulsed wave Doppler echocardiography evaluates simultaneous flow across the mitral and aortic valves or the superior vena cava and aorta to demonstrate the lack of association between the atrial and ventricular flows [11]. Fetal magnetocardiography (fMCG), while not widely available, is more precise in defining the type of fetal arrhythmia and the mechanism of conduction [12,13]. In our patient, fMCG was used to confirm the diagnosis of CCHB with a low fetal ventricular rate of 38–41 bpm at 24 weeks gestation. fMCG may also contribute to significant changes in diagnosis and management in these cases, including additional medical therapy, closer surveillance interval, and delivery planning [14].

Our patient also had associated complications of significant fetal ascites at the time of diagnosis. There have been rare reports of ascites that did not appear attributable to congestive heart failure in the setting of CCHB, which resolved with maternal steroid administration [15]. We suspect the fetal ascites was also immune-mediated in our case, given that this abnormal fluid accumulation was documented prior to development of CCHB. However, the ascites in our case did not resolve with steroid therapy, and additional signs of hydrops developed as the pregnancy progressed, with a steadily declining CVP.

### 3.3. Prenatal Management

There continues to be no standard medical therapy for the treatment of CCHB, and existing therapies have variable maternal side effects. Fluorinated steroids such as dexamethasone are frequently used to reduce inflammation [12]. Some reports have shown that these may reverse or stabilize incomplete atrioventricular block, though this is not seen consistently [6,13,16,17]. Side effects may include maternal glucose intolerance, oligohydramnios, and transient fetal hypoadrenalism. Our patient received dexamethasone at diagnosis of intermittent second-degree heart block prior to developing complete heart block. However, there was no stabilization of the fetal heart rhythm or improvement in ascites noted in our patient. Intravenous immunoglobulin can also be considered to decrease the volume of circulating maternal antibodies [6,12]. Beta sympathomimetics, such as terbutaline, can increase fetal ventricular rates and are generally only considered when the ventricular rate is below 55 bpm [6,11,13]. Terbutaline has been well-tolerated by mothers in several studies, with only mild maternal tachycardia and benign ectopy reported [12]. Terbutaline typically produces a more significant increase in the atrial and ventricular rates in isolated CCHB versus CCHB associated with left atrial isomerism [4]. Our patient was offered terbutaline therapy due to the extremely low fetal ventricular rate, but she declined additional treatment. Fetal pacing is another potential therapy that has primarily been tried investigationally for cases refractory to transplacental medical therapy, with variable success. The advent of micro pacemaker devices may further this therapy in the future [11].

Given the lack of standardized medical therapy for treatment of CCHB, using a shared decision-making model for any prenatal interventions is critical, as we consider the impact not only for the fetus but for the mother as well. The goals of the mother and her support system may shift throughout the pregnancy, and these conversations about treatment possibilities should occur at multiple visits as a part of the general prenatal counseling to ensure these goals are being met appropriately throughout gestation.

### 3.4. Delivery Room and Post-Natal Management

Delivery room and neonatal resuscitation planning is critical in cases of CCHB, given the risk of hemodynamic instability from bradycardia-induced poor systemic perfusion. In many cases, support with acute medical and surgical interventions is necessary to increase heart rate and cardiac output. The goal is to minimize oxygen consumption with ventilation and sedation/neuromuscular blockade, increase neonatal heart rate, and support systemic perfusion during transport from the delivery room to the cardiac surgical unit. This may include emergent temporary pacing with subsequent placement of a permanent pacemaker. To facilitate these urgent interventions, delivery should be planned at a tertiary care center where pediatric cardiology, pediatric electrophysiology, and cardiothoracic surgery specialties are readily available [18]. Having all necessary team members, medications, and devices ready to provide interventions at the time of delivery is essential for neonatal survival. Pre-term delivery may be warranted if there is evidence of fetal distress or deteriorating cardiac status [19]. Signs of cardiac distress can be assessed serially using the CVP with scores of 6–7 or lower indicating a higher risk of perinatal mortality [20]. Our patient’s CVP was 6 at diagnosis and decreased to 4 by the final visit, prompting discussions for early delivery.

The mainstay of postnatal treatment is permanent pacemaker insertion for those who are symptomatic, have ventricular dysfunction, or meet specific electrocardiographic criteria including wide complex QRS, complex ventricular ectopy, or rate < 55 bpm [5]. Permanent pacing is eventually required in most cases of CCHB. In one study, 53% of neonates required pacing, with an additional 40% requiring pacing at an older age [7].

Primary prevention strategies to prevent the development of CCHB have also been evaluated. For example, in mothers with autoimmune diseases and circulating antibodies, using hydroxychloroquine therapy starting from early gestation has shown promising results in decreasing the likelihood of developing CCHB [11].

## 4. Conclusions

CCHB is a rare disease that can be diagnosed in utero but has limited definitive fetal treatment options. Outcomes remain guarded, with significant morbidity and mortality reported throughout the literature during fetal and postnatal life. Extremely low fetal ventricular rates increase the likelihood of poorer outcomes, as occurred in our case with a low fetal ventricular rate of 25 bpm. Prenatal counseling on treatment options and outcomes should include shared decision-making to maximize goals of care being met.

## Figures and Tables

**Figure 1 children-10-01132-f001:**
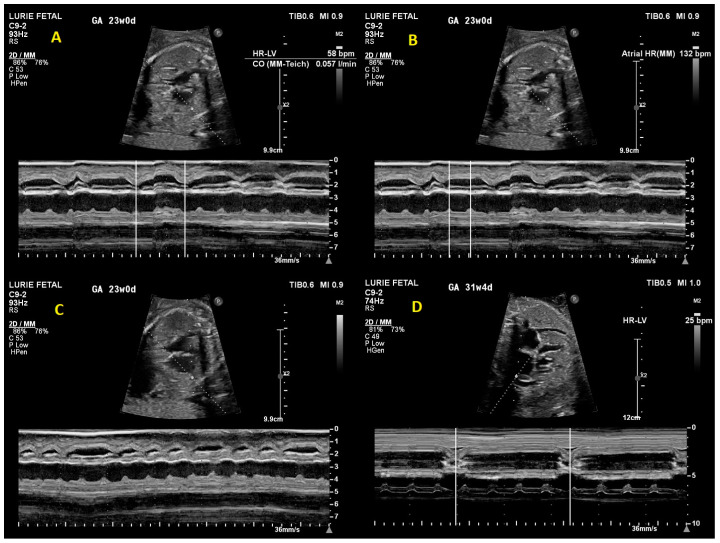
M-mode tracings from initial fetal echocardiogram demonstrating (**A**) ventricular rate of 58 bpm, (**B**) atrial rate of 132 bpm, and (**C**) second degree heart block. (**D**) M-mode tracing from final fetal echocardiogram demonstrating ventricular rate of 25 bpm with complete heart block.

**Figure 2 children-10-01132-f002:**
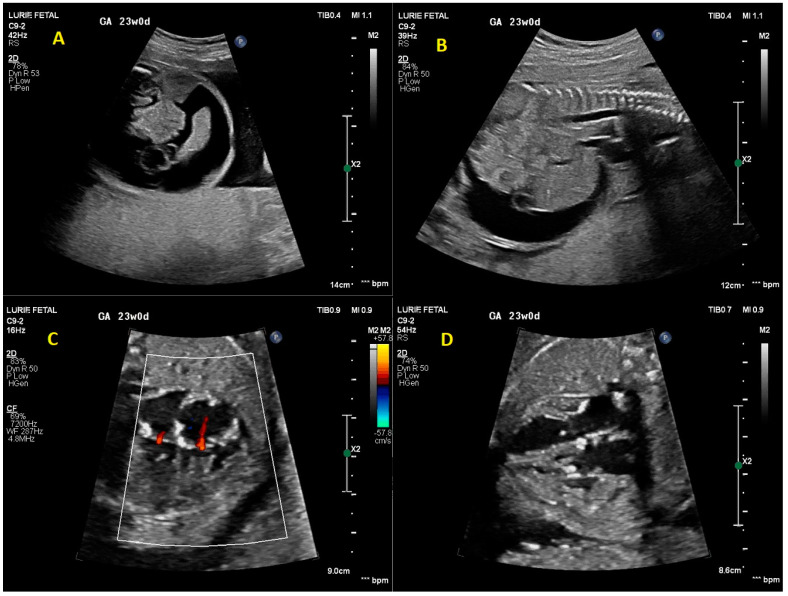
Initial fetal echocardiogram findings. (**A**) Significant ascites (axial plane). (**B**) Significant ascites (sagittal plane). (**C**) Mild tricuspid and mitral valve regurgitation. (**D**) Increased echogenicity of papillary muscles, AV valves, and interatrial septum.

**Figure 3 children-10-01132-f003:**
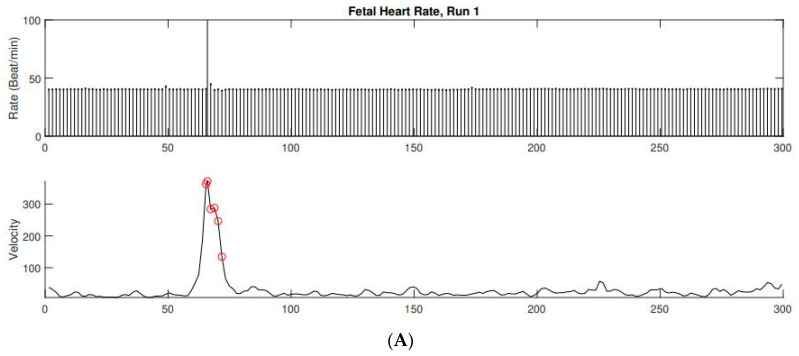
(**A**) FMCG-derived fetal actocardiogram showing fetal heart rate in the upper tracing and fetal movement in the lower tracing. The fetal heart rate was non-reactive but fetal movement was associated with ectopy. The baseline fetal heart rate was 40 bpm. (**B**) FMCG tracing showing AV dissociation with an atrial rate of 131 bpm. The QRS duration (82 ms) is indicative of a wide escape rhythm.

## Data Availability

No new data were created or analyzed in this study. Data sharing is not applicable to this article.

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
