# Peer review of "Fetal Congenital Complete Heart Block: A Rare Case with an Extremely Low Ventricular Rate and Review of Current Management Strategies"

_children, 2023, doi:10.3390/children10071132_

Round 1
Reviewer 1 Report
The paper „Fetal congenital complete heart block: a rare case with an extremely low ventricular rate and review of current management strategies” concerns very rare but clinically significant phenomenon where the degree of fetal atrioventricular block deteriorates in pregnant woman with autoimmune disease. The authors precisely described whole process with accurate echocardiographic images. The authors nicely presented applied treatment and conducted decission-making process in regard to fetal fate. In the section of discussion the authors thoroughly analyzed available epidemiological data, pathophysiology of congenital complete heart block and it’s possible treatment methods.
If only the paper doesn’t exceed word count required by the Journal, it can be published without any revision.
Author Response
Thank you for the feedback
Reviewer 2 Report
Congratulations, despite poor outcome we need to report also about our failures !!!!
Author Response
Thank you for the feedback. These cases often have poor outcomes which we agree is relevant to present as well.
Reviewer 3 Report
The authors present a case report of a fetus with an isolated complete heart block and the lowest reported fetal heart rate.
The case is well established with sufficient documentation of the heart failure progression. It includes adequate echocardiography and fetal magnetocardiography images demonstrating the severity of the disease.
As mentioned, there is no addressed fetal therapy. While the family refused every possible supportive therapeutic option, the fetal demise was as expected.
It also emphasizes the role and problems of prenatal counseling.
I don't understand the possible competition for the lowest documented fetal heart rate, especially when the fetus has died. But the case report is very nicely done and is worth publishing.
Author Response
Thank you for the feedback. Wording was added to stress that the fetal heart rate can remain low for a prolonged time period which may allow additional time for treatment or consideration of early delivery as in our case. While the patient and family in our case declined, this is an important management consideration overall.